# Circulating Levels of Nesfatin-1 and Spexin in Children with Prader-Willi Syndrome during Growth Hormone Treatment and Dietary Intervention

**DOI:** 10.3390/nu15051240

**Published:** 2023-03-01

**Authors:** Joanna Gajewska, Katarzyna Szamotulska, Witold Klemarczyk, Magdalena Chełchowska, Małgorzata Strucińska, Jadwiga Ambroszkiewicz

**Affiliations:** 1Department of Screening Tests and Metabolic Diagnostics, Institute of Mother and Child, 01-211 Warsaw, Poland; 2Department of Epidemiology and Biostatistics, Institute of Mother and Child, 01-211 Warsaw, Poland; 3Department of Nutrition, Institute of Mother and Child, 01-211 Warsaw, Poland

**Keywords:** Prader-Willi syndrome, anorexigenic peptides, nesfatin-1, spexin, leptin, adiponectin, children

## Abstract

Background: Despite observable improvement in the treatment outcomes of patients with Prader-Willi syndrome (PWS), adequate weight control is still a clinical problem. Therefore, the aim of this study was to analyze the profiles of neuroendocrine peptides regulating appetite—mainly nesfatin-1 and spexin—in children with PWS undergoing growth hormone treatment and reduced energy intake. Methods: Twenty-five non-obese children (aged 2–12 years) with PWS and 30 healthy children of the same age following an unrestricted age-appropriate diet were examined. Serum concentrations of nesfatin-1, spexin, leptin, leptin receptor, total adiponectin, high molecular weight adiponectin, proinsulin, insulin-like growth factor-I, and total and functional IGF-binding protein-3 concentrations were determined using immunoenzymatic methods. Results: The daily energy intake in children with PWS was lower by about 30% (*p* < 0.001) compared with the controls. Daily protein intake was similar in both groups, but carbohydrate and fat intakes were significantly lower in the patient group than the controls (*p* < 0.001). Similar values for nesfatin-1 in the PWS subgroup with BMI Z-score < −0.5 and the control group, while higher values in the PWS subgroup with BMI Z-score ≥ −0.5 (*p* < 0.001) were found. Spexin concentrations were significantly lower in both subgroups with PWS than the controls (*p* < 0.001; *p* = 0.005). Significant differences in the lipid profile between the PWS subgroups and the controls were also observed. Nesfatin-1 and leptin were positively related with BMI (*p* = 0.018; *p* = 0.001, respectively) and BMI Z-score (*p* = 0.031; *p* = 0.027, respectively) in the whole group with PWS. Both neuropeptides also correlated positively in these patients (*p* = 0.042). Conclusions: Altered profiles of anorexigenic peptides—especially nesfatin-1 and spexin—in non-obese children with Prader-Willi syndrome during growth hormone treatment and reduced energy intake were found. These differences may play a role in the etiology of metabolic disorders in Prader-Willi syndrome despite the applied therapy.

## 1. Introduction

Prader-Willi syndrome (PWS) is a rare congenital neurodevelopmental disorder characterized by hyperphagia and many behavioral disturbances leading to morbid obesity [1]. In addition, hypotonia in early infancy, short stature, hypogonadism, and developmental delay are observed in patients with this syndrome. PWS is caused by the loss of genes on the paternally acquired chromosome 15q11-q13 and the prevalence of this syndrome is 1/10,000–1/30,000 cases worldwide. The conservation of the PWS genetic interval on human chromosome 15q11-q13 and the gene cluster on mouse chromosome 7 facilitated the use of mice as animal models of PWS [2]. Some models mimicked the loss of all gene expression from the paternally inherited PWS genetic interval, while others considered smaller regions or single genes. These models revealed a number of mechanisms responsible for hypothalamic dysfunctions, resulting in hyperphagia, growth retardation, and metabolic disorders. Most of the typical features of PWS may be due to hypothalamic dysfunction both in orexigenic and anorexigenic neural populations [3]. Abnormal brain networks disrupt the physiological control of food intake and weight, which is observed in patients with PWS [4]. There are a number of nutritional phases in children and adolescents with PWS [5]. During phase 1, the infants are hypotonic and not obese. Phase 2 is associated with weight gain, but in sub-phase 2a the weight increases without a significant change in appetite or caloric intake (age quartiles 20–31 months). Next, in sub-phase 2b the weight gain is associated with a concomitant increased interest in food (quartiles 3–5.25 years). Phase 3 is characterized by hyperphagia accompanied by a lack of satiety (quartiles 5–13 years). Among the appetite-regulating neuroendocrine factors, there are orexigenic factors that stimulate food intake and/or increase body weight and anorexigenic factors that inhibit food intake and/or decrease in body weight [6]. Leptin—an anorexigenic peptide mainly secreted by white adipose tissue—reduces food intake and energy metabolism at the hypothalamus level via melanocortin receptors [7]. Soluble leptin receptor (sObR) is the main leptin-binding protein in the blood and can affect leptin bioavailability and functioning [8]. Plasma leptin in patients with PWS is positively correlated with body mass index (BMI) and body fat mass, but lower, unchanged as well as higher, leptin concentrations were found in these patients in comparison with healthy controls [9,10].

Adiponectin is another peptide secreted by adipocytes, but its effect on feeding behavior is controversial and closely related to nutritional status and food consumption [11]. Higher adiponectin concentrations were found in patients with PWS compared with obese and non-obese controls [12,13], whereas other authors observed no differences compared with obese subjects [14].

Nesfatin-1 and spexin were recently identified anorexigenic peptides involved in energy homeostasis that have not yet been studied in PWS patients. Nesfatin-1 is produced from nucleobindin-2 (NUCB2) precursor protein [15]. As a result of the post-translational modification of NUCB2, three cleavage products are formed: nesfatin-1 (amino acids 1–82), nesfatin-2 (amino acids 85–163), and nesfatin-3 (amino acids 166–396). Among these three forms, only nesfatin-1 shows biological activity. The physiological effect of nesfatin-1 relates to the reduction of food intake through central and peripheral actions. This peptide is expressed in the central nervous system (CNS) and peripheral tissues, such as adipose tissue, gonads, stomach, pancreas, and liver [16]. To date, the mechanism of nesfatin-1 action has not been clarified. This peptide may act in the inhibition of feeding via oxytocin, melancortin, and other systems to relay its anorexigenic properties [17]. Several studies showed that peripheral nesfatin-1 may be associated with BMI and body fat in obese children and adults, however, positive as well as negative correlations between BMI and serum nesfatin-1 levels were found in obese patients [18,19,20]. 

Spexin (SPX) is a 14 amino acid peptide encoded by the C12orf29 gene and located on chromosome 12 of the human genome [21]. This peptide is mainly produced in white adipose tissue, brain, heart, muscles, ovaries, testes, and gastrointestinal tract [22]. Spexin plays a role in glucose and lipid metabolism. Thus, it is speculated that this peptide may be one of the protective agents against obesity and metabolic syndrome (MetS) [23]. Some studies reported no differences in spexin concentrations between obese and normal-weight subjects, whereas other studies obtained lower values of spexin in obese subjects compared with non-obese controls [24,25]. It has been suggested that altered spexin expression may affect the cross talk between the brain and peripheral organs in obese patients [26].

Current therapeutic strategies in PWS consist of preventive methods for weight management and mainly include early therapy with growth hormone (GH), dietary recommendations, and behavioral interventions [1]. Despite observable improvement in the treatment outcomes of patients with PWS, adequate weight control is still a clinical problem as patients often develop obesity and/or metabolic disorders [27]. Therefore, the aim of this study was: (i) to analyze the differences in the profiles of circulating peptides regulating appetite—mainly nesfatin-1 and spexin—in children with PWS undergoing GH treatment and reduced energy intake with the profiles in healthy, non-obese children, and (ii) to evaluate the relationships between the biochemical parameters and anthropometric indices in children with PWS during GH treatment and dietary intervention.

## 2. Materials and Methods

### 2.1. Patients

We examined 25 Caucasian children with Prader-Willi syndrome aged 2–12 years, who were recruited between 2020 and 2022 from a group of consecutive patients seeking dietary counseling in the Department of Nutrition at the Institute of Mother and Child in Warsaw, Poland. The inclusion criteria for this study were: (a) genetically confirmed diagnosis of PWS; (b) GH treatment for at least one year, and being on GH at the time of inclusion (0.025 mg/kg/day); (c) being on an energy-restricted diet. The exclusion criteria were: (a) a body mass index (BMI) Z-score > 1; (b) chronic secondary illness, such as diabetes mellitus, liver, or kidney disease; (c) taking investigational drugs; (d) not signing the informed consent form. The average duration of treatment with growth hormone in the whole group of patients with PWS was 4.7 ± 2.8 years. 

The control group consisted of 30 non-obese, healthy children (BMI Z-score < −1 + 1) within the same age range as the group with PWS, an adequate nutritional or dietary status according to the recommendations of Kułaga et al. [28] and Jarosz et al. [29]. The control group in the study were children: (a) without acute or chronic disorders; (b) not taking any medications that could affect their development and nutritional or dietary status; (c) IGF-1 levels within the normal range for age and sex. Written informed consent was obtained from the parents of all the examined children. The study was performed in accordance with the Helsinki Declaration for Human Research, and the study protocol was approved (protocol code: 8/2020; date of approval: 6 April 2020) by the Ethics Committee of the Institute of Mother and Child in Warsaw, Poland.

### 2.2. Assessment of Dietary Intake

Based on lower energy requirements for children with PWS, the recommendations included limiting caloric intake by 20–40% along with a well-balanced macronutrient distribution [30]. Two weeks before the child was due to visit the Department of Nutrition, a food diary was completed by the parents at home. The parents had previously been trained by a nutritionist to provide estimates of diary intake. Next, nutritionists carried out interviews concerning nutritional behaviors, and checked the diary in the presence of the child and their parents. The nutritionist also asked for detailed information about the recorded foods and drinks, such as portion sizes and preparation methods. The portion sizes were corrected during the visit using a photo album of products and dishes presenting meal portion sizes [31]. The three-day methodology was used according to the methodological guide on nutrition research to assess intake in the children’s dietary habits [32]. The data of the three-day dietary records—two weekdays and one weekend day—were entered into the nutrition analysis software (Dieta 5^®^, National Food and Nutrition Institute, Warsaw, Poland) to assess the average daily energy intake and the percentage of energy intake from fat, protein and carbohydrates [33]. The data for each participant were compared with the recommendations for the appropriate age and sex. The age- and sex-specific percentage of estimated energy requirement (EER) for total energy intake and adequate intake (AI) for fiber were calculated [29]. The children in the present study did not receive supplements, except standard supplementation with vitamin D.

### 2.3. Anthropometric Measurements

Physical examinations—including body height and weight measurements—were performed in both of the studied groups. Body height was measured using a standing stadiometer and recorded with a precision of 1 mm. Body weight was assessed unclothed —to the nearest 0.1 kg—with a calibrated balance scale. Body mass index(BMI, body weight divided by height squared, kg/m^2^) of each individual was converted to BMI Z-score for the child’s age and sex using Polish reference tables [28]. Body composition was measured by dual-energy X-ray absorptiometry (DXA) using Lunar Prodigy (General Electric Healthcare, Madison, WI, USA) using pediatric software version 9.30.044. All children were measured with the same equipment using standard positioning techniques.

### 2.4. Biochemical Analyses

Venous blood samples were collected between 8:00 and 10:00 a.m. after an overnight fast. To obtain serum, the blood was centrifuged at 1000× *g* for 10 min at 4 °C. Serum specimens were stored at −70 °C prior to assay. Serum nesfatin-1, spexin, leptin, leptin receptor, total adiponectin, high molecular weight (HMW) adiponectin, proinsulin, insulin-like growth factor-I (IGF-I), total-IGF-binding protein-3 (t-IGFBP-3), and functional IGFBP-3 (f-IGFBP-3) concentrations were determined using immunoenzymatic methods. The concentrations of nesfatin-1 and spexin were determined by the Elisa kits (Elabscience, Houston, TX, USA) with anti-human NES-1 antibody and anti-human SPX antibody, respectively. The intra- and inter-assay CVs were 4.8% and 5.2% for nesfatin-1, and 5.1% and 4.2% for spexin, respectively.

Elisa kits from DRG Diagnostics (Marburg, Germany) were used to determine leptin and soluble leptin receptor (sOB-R) concentrations. The intra- and inter–assay CVs were less than 9.6% and 9.1% for leptin, and 7.2% and 9.8% for leptin receptor, respectively. 

Serum levels of total adiponectin and HMW adiponectin were determined using an ELISA kit (ALPCO Diagnostics, Salem, NH, USA). The intra- and inter-assay variations were less than 5.7% and 6.4%, respectively. The concentrations of proinsulin were measured using kit from TECO Medical (Sissach, Switzerland). The intra- and inter-assay variations were less than 2.2% and 4.0%, respectively.

IGF-I and t-IGFBP-3 values were determined using ELISA kits from Mediagnost (Reutlingen, Germany). The intra- and inter-assay coefficients of variation were less than 6.7% and 6.6% for IGF-I, and 2.2% and 7.4% for t-IGFBP-3, respectively. We calculated the IGF-I/IGFBP-3 molar ratio—an estimation of free IGF-I concentration—as [IGF-I (ng/mL) × 0.130]/[IGFBP-3 (ng/mL) × 0.036]. The IGF-I/IGFBP-3 molar ratio has been used as a surrogate parameter associated with IGF-I bioactivity [25]. 

The f-IGFBP-3 concentration was determined using ligand-binding immunoassay (LIA) (Mediagnost, Reutlingen, Germany) with intra- and inter-assay variability of less than 5.6% and 6.8%, respectively. This assay for f-IGFBP-3 exclusively detects IGFBP-3 capable of IGF binding. The f-IGFBP-3/t-IGFBP-3 molar ratio was calculated to serve as an index of IGFBP-3 fragmentation. A ratio close to 0 indicates almost complete fragmentation and loss of IGFBP-3 function, whereas a ratio close to 1 indicates the presence of mostly intact protein with unchanged biological function. The analysis of each parameter was performed in duplicate.

Total cholesterol, LDL and HDL-cholesterol, triglycerides and glucose levels were measured using standard methods (Roche Diagnostics, Basel, Switzerland). 

### 2.5. Statistical Analyses

The results are presented as means ± standard deviation (SD) for symmetric distributed data or medians and interquartile range (25–75th percentiles) for skewed distributed variables. The Kolmogorov-Smirnov test was used to evaluate distribution for normality. Differences in anthropometric characteristics, biochemical parameters, and dietary intake of patients with Prader-Willi syndrome and healthy, non-obese children were assessed using the non-parametric Mann-Whitney *U* test. 

We categorized children with PWS and the control group into subgroups according to BMI Z-score: PWS 1 (n = 13) and Controls 1 (n = 10) with BMI Z-score < −0.5; PWS 2 (n = 12) and Controls 2 (n = 20) with BMI Z-score ≥ −0.5. Differences in BMI, fat mass, biochemical parameters—including nesfatin-1, spexin, leptin, and adiponectin—and dietary intake were compared between the subgroups with PWS with the appropriate control subgroups.

Quantile regression was used to assess the age-adjusted relationships between the studied appetite-regulating peptides as well as the age-adjusted relationships between appetite-regulating peptides and selected anthropometric parameters in the entire group of patients with PWS. For trend analysis, the Jonckheere-Terpstra test was used. A *p*-value < 0.05 was considered to be statistically significant. Statistical analysis was performed using IBM SPSS v.25.0 software (SPSS Inc., Chicago, IL, USA) and Stata Statistical Software: Release 17 (StataCorp. 2021, College Station, TX, USA: StataCorp LLC).

## 3. Results

### 3.1. Clinical Characteristics and Dietary Intake of Children with PWS

Significantly lower BMI and BMI Z-score values were observed in children with PWS than healthy children of the same age (*p* < 0.05), but a similar body fat mass was found in both groups (Table 1). 

Children with PWS had higher serum nesfatin-1 levels (*p* = 0.019) by 40% and lower serum spexin concentrations by half (*p* < 0.001) compared with the controls. Significant differences in lipid profile between these groups were also observed. Higher concentrations of total cholesterol by about 15% (*p* = 0.001), LDL cholesterol by about 25% (*p* < 0.001), and triglycerides by about 25% (*p* = 0.005) were found in patients with PWS than in healthy children. Similar values of serum leptin, sOB-R, total adiponectin, HMW adiponectin, and glucose were found in both studied groups. However, slightly higher proinsulin concentrations (*p* = 0.069) were observed in children with PWS.

The IGF-I, t-IGFBP-3 concentrations, and the IGF-I/t-IGFBP-3 molar ratio in patients with PWS were higher by 2-fold (*p* < 0.001), 20% (*p* = 0.016), and 50% (*p* < 0.001) than in controls, respectively. The level of functional IGFBP-3 in patients was higher by about 30% (*p* = 0.055) than in healthy children, but the *p*-value was borderline. The f-IGFBP-3/t-IGFBP-3 molar ratio was similar in both studied groups.

The daily energy intake and the percentage of EER in children with PWS were lower (*p* < 0.001), but the percentage of energy from proteins was significantly higher (*p* < 0.001) than in healthy children (Table 2). 

The proportion of carbohydrates in daily energy intake was similar in both groups, but the proportion of fat was significantly lower in patients with PWS than controls (*p* = 0.018). Daily protein intake was similar in both groups, but daily carbohydrate and fat intakes were significantly lower in the group with PWS than in the controls (*p* = 0.001). Lower daily intake in patients with PWS was also found for cholesterol (*p* = 0.006) and saturated fatty acids (*p* = 0.001). Similar values for fiber intake were observed in both groups. 

### 3.2. Biochemical Characteristics and Dietary Intake of the PWS Subgroups with Lower BMI Z-Score (BMI Z-Score < −0.5) and Higher BMI Z-Score (BMI Z-Score ≥ −0.5)

Table 3 shows the comparison of the PWS subgroups with lower BMI Z-score (PWS 1; BMI Z-score < −0.5) and higher BMI Z-score (PWS 2; BMI Z-score ≥ −0.5) with the control subgroups.

No differences were observed concerning age, height, BMI, and fat mass between the subgroups with PWS and the respective healthy children, except a lower BMI Z-score in PWS 2 than Controls 2 (*p* = 0.035). 

The PWS 2 subgroup with higher BMI Z-score was characterized by higher BMI (*p* < 0.001) and fat mass percentage (*p* = 0.031), higher leptin/sOB-R and leptin/adiponectin ratios (*p* < 0.001) and lower sOB-R concentrations (*p* = 0.016) than the PWS 1 subgroup with lower BMI Z-score. We observed similar values of lipid parameters in PWS 1 and PWS 2 subgroups (*p* > 0.05), but when the PWS subgroups were compared with the corresponding controls, significantly higher concentrations of total cholesterol, LDL-cholesterol, and triglycerides were found in patients with PWS. The concentrations of sOB-R, total adiponectin, HMW adiponectin, proinsulin, and glucose were similar in patient and control subgroups (*p* > 0.05).

The daily energy intake was lower (*p* = 0.004) in PWS 1 compared with Controls 1, but the percentage of energy from protein was significantly higher (*p* = 0.030) than in healthy children. The proportion of carbohydrates in daily energy intake was similar in both subgroups, but the percentage of energy from fat was significantly lower than in Controls 1 (*p* = 0.049). Lower daily intake in PWS 1 was also found for fat (*p* = 0.001), cholesterol (*p* = 0.015), and saturated fatty acids (*p* = 0.004). Fiber intake was higher in PWS 1 than in Controls 1. 

Daily intake did not differ statistically significantly, but the percentage of EER in children in PWS 2 was lower (*p* < 0.001) compared with Controls 2. The percentage of energy from protein was significantly higher (*p* < 0.001) than in healthy children, but the proportions of carbohydrates and fat in daily energy intake were similar in both groups. Daily protein and fat intake were similar in both groups, but daily carbohydrate intake was significantly lower in PWS 2 than in Controls 2 (*p* = 0.009). Lower daily intake in PWS 2 was also found for saturated fatty acids (*p* = 0.033). Similar values for fiber intake were observed in both subgroups. 

### 3.3. Appetite-Regulating Peptides in the PWS Subgroups with Lower BMI Z-Score (BMI Z-Score < −0.5) and Higher BMI Z-Score (BMI Z-Score ≥ −0.5)

Analyzing neuropeptide concentrations in the PWS subgroups compared with the controls, we observed similar values for nesfatin-1 in PWS1 and Control 1 (*p* = 0.446), but higher in PWS 2 than in Control 2 (*p* < 0.001) (Figure 1A). Moreover, nesfatin-1 values were higher in PWS 2 than PWS 1 (*p* < 0.001).

Spexin concentrations were significantly lower in PWS subgroups and controls (PWS 1 vs. Controls 1, *p* < 0.001; PWS 2 vs. Controls 2, *p* = 0.005) (Figure 1B). In fact, spexin concentrations were higher in the PWS 2 subgroup than PWS 1, but the *p*-value was borderline (*p* = 0.054).

Leptin and adiponectin concentrations were similar in PWS compared with the Control subgroups (Figure 1C,D). However, when comparing both PWS subgroups, leptin concentrations were higher in PWS 2 than PWS 1 (*p* = 0.001).

### 3.4. Associations between Peptides Regulating Appetite and Anthropometric Parameters in Children with PWS

Age-adjusted associations between peptides regulating appetite and between these peptides and anthropometric parameters in the entire group of children with PWS are presented in Table 4. 

Nesfatin-1 and leptin were associated positively with BMI (*p* = 0.018; *p* = 0.001, respectively), BMI Z-score (*p* = 0.031; *p* = 0.027, respectively), and leptin additionally with body fat mass percentage (*p* = 0.026). Moreover, both peptides were associated positively in patients with PWS (*p* = 0.042). We did not observe any associations between spexin and adiponectin with anthropometric and biochemical parameters in these patients. In addition, we did not observe any associations between peptides regulating appetite and other anthropometric and biochemical parameters in patients with PWS (*p* > 0.05).

### 3.5. Nesfatin-1 and Spexin Concentrations in Children with PWS Depending on the Nutritional Phase

In children with PWS, we found significant associations between both neuropeptides and the nutritional phases (*p*_trend_ = 0.004 for nesfatin-1; *p*_trend_ = 0.041 for spexin) (Table 5). 

We observed the highest values of these peptides in the group with PWS aged 6–12 years (phase 3). In healthy children, we did not find any associations between the concentrations of nesfatin-1 and spexin and the nutritional phases.

## 4. Discussion

Obesity in Prader-Willi syndrome resulting from hyperphagia can be prevented by caloric intake restriction [34]. To maintain a healthy weight and avoid obesity, children with PWS required about 30% less energy in the presented study. Moreover, these patients consumed less fat, less carbohydrates, a similar amount of protein, and a fairly high amount of fiber. This is in line with many studies showing that children with PWS require a 20–40% reduction in energy intake to maintain a healthy body weight [30]. In our study, all children with PWS were not obese and even had slightly lower BMI and BMI Z-score values and a similar amount of body fat mass compared with healthy children. Besides a low-energy diet, the BMI values of our patients could also be modulated by GH treatment. Irizarry et al. [35] found that GH treatment was associated with a lower BMI Z-score and higher IGF-I in patients with PWS. We also observed higher concentrations of IGF-I and t-IGFBP3, and slightly higher f-IGFBP3 concentrations in children with PWS than in the controls. The f-IGFBP-3/t-IGFBP-3 molar ratio obtained in our study was similar in both groups and reflected the same degree of this protein fragmentation in PWS and healthy children. The IGF-I/IGFBP-3 molar ratio as an indicator of free IGF-I, with a higher ratio in our patients due to excess IGF-I may reflect higher IGF-I bioactivity. According to some authors, young children with PWS treated with GH would need relatively high IGF bioactivity because they grow fast [36]. 

Scientific research on the treatment of patients with PWS attributes an important role to nutritional intervention to ensure sustainable growth while preventing obesity and malnutrition in these patients. The exact mechanism of obesity development in Prader-Willi syndrome is not fully understood. Abnormalities in the hypothalamic satiety center and its hormonal circuits can affect energy expenditure, food intake, body composition, and endocrine factors deficiencies in patients with PWS [3]. 

The present study is the first to evaluate anorexigenic neuropeptides—nesfatin-1 and spexin—levels in relation with anthropometric parameters and other peptides regulating appetite in children with PWS. In our study, we found higher concentrations of nesfatin-1 in children with PWS than in healthy children and positive associations between nesfatin-1 and BMI and BMI Z-score. In addition, higher values of this peptide were observed in patients with the higher BMI Z-score (BMI Z-score ≥ −0.5) than with the lower BMI Z-score (BMI Z-score < −0.5). It seems that the higher nesfatin-1 concentrations may appear in response to the body’s energy status and/or nutritional phase. Patients belonging to the age group characterized by hyperphagia (phase 3) had the highest values of nesfatin-1 in comparison with nutritional phases 2a and 2b. In addition, patients with a higher BMI Z-score were characterized not only by higher nesfatin-1 concentrations, but also by higher adipose tissue mass, higher leptin concentrations, and a higher leptin/sOB-R ratio than children with lower BMI-Z-scores. We also found positive associations between nesfatin-1 and leptin in children with PWS. 

Conflicting results were reported regarding fasting serum nesfatin-1 concentrations and the relation between nesfatin-1 and BMI values in malnourished as well as obese children [18,19,20,37,38,39]. Lower nesfatin-1 levels in acute malnourished children were found by Kahraman et al. [39] and higher levels of this adipokine were observed by Kaba et al. [37] and Acar et al. [38]. The authors suggested that high nesfatin-1 may be one of the reasons for chronic malnutrition by causing poor appetite due to feelings of satiety. However, a positive [39], negative [38], and no correlations [37] between BMI SDS and nesfatin-1 were observed in children with malnutrition. It also cannot be ruled out that nesfatin-1 synthesis and serum nesfatin-1 concentration may differ in the studied populations due to nesfatin-1 gene polymorphism, as suggested by some authors [38,40].

Different results regarding serum nesfatin-1 concentrations were observed in obese than non-obese children by other authors [18,19,20,41]. Abaci et al. [19] and Kim et al. [20] reported lower serum nesfatin-1 levels in obese subjects compared with healthy controls and a negative correlation between nesfatin-1 and BMI in these patients. The authors speculated that low levels of this satiety peptide may be one of the reasons for inadequately controlled food intake. However, Anwar et al. [18] found higher serum nesfatin-1 in obese children (range 5–15 years) than in control subjects and positive correlations between nesfatin-1 and BMI-SDS. According to Tan et al. [41] the saturation of transporters for nesfatin-1 into cerebrospinal fluid (CSF) may explain the higher concentrations of this peptide in plasma.

Nesfatin-1 exerts anorexigenic functions, but according to Dore et al. [42] it works independently of the leptin pathway. However, nesfatin-1 signaling may be important in mediating leptin-induced anorexia [43]. Wernecke et al. [44] suggest common downstream signaling mechanisms for both peptides, because central co-administration of leptin and nesfatin-1 did not yield larger effects on energy expenditure than nesfatin-1 or leptin alone. The results obtained in our study also do not exclude the presence of functional relationships between nesfatin-1 and leptin in Prader-Willi syndrome. 

Spexin, the next anorexigenic peptide, was significantly lower in our children with PWS compared with healthy children. We did not observe any relations between spexin and other biochemical and anthropometric parameters. Circulating spexin levels were also lower in obese children compared with normal-weight controls and did not correlate with other adipokines and cardiometabolic risk factors [45,46]. However, we found significant associations between spexin and the nutritional phases in our patients with PWS. Spexin signals neurons in the hypothalamus directly reducing food intake by enhancing leptin receptor and melanocortin 4 receptor expression, while decreasing neuropeptide Y type 5 receptor and ghrelin receptor expression [47]. Therefore, it is suggested that spexin expression is regulated by metabolic status or feeding conditions.

The role of spexin in obesity appears to be related to various factors, such as the regulation of appetite, eating behaviors, regulation of body weight, glucose homeostasis, and inhibiting long-chain fatty acid uptake into adipocytes [48]. This neuropeptide regulates fat tissue metabolism through the induction of lipolysis and inhibition of lipogenesis in human adipocytes and murine 3 T3-L1 cells [49]. Spexin has been found to efficiently reduce total lipids in the liver, suggesting its key role in lipid metabolism in mammals [50]. In our non-obese patients with PWS, lower concentrations of spexin were observed with altered lipid profile. Kavalahat et al. [26] reported that circulating levels of spexin were decreased with obesity and diabetes in adults and inversely correlated with lipid markers such as total cholesterol, LDL cholesterol and triglycerides, but positively correlated with HDL levels. Although we did not observe any significant associations between spexin and lipids, the influence of the deficit of this peptide on lipid metabolism in children with Prader-Willi syndrome may be considered. 

Despite growth hormone therapy, metabolic disorders such as dyslipidemia and insulin resistance are described in some patients with PWS [27]. It is known that adiponectin is associated with increased insulin sensitivity and has anti-inflammatory properties [51]. In our patients—compared with controls similar—concentrations of adiponectin, HMW adiponectin, glucose, and slightly higher proinsulin concentrations were observed. Other authors found higher plasma concentrations of this adipokine in the PWS population, which was correlated with insulin sensitivity in these patients [52]. 

The present study had several limitations. First, we had a relatively small number of participants owing to the rarity of Prader-Willi syndrome. However, the study group was homogeneous in terms of therapy (GH therapy and low-energy diet) and anthropometrically and biochemically. The second limitation of this study was its cross-sectional nature and the absence of a prospective longitudinal analysis, which is needed to examine the relationship between circulating peptides regulating appetite and clinical outcomes in these subjects during therapy. Therefore, the assessment of the clinical utility of nesfatin-1 and spexin in patients with PWS requires long-term monitoring of the concentrations of these peptides during growth hormone therapy as well as dietary intervention. The next limitation was the lack of comparison between non-obese and obese children with PWS. In our Institute, early diagnosis and therapeutic intervention significantly reduce the number of obese patients with PWS. Further studies are needed to clarify the functional relationships between anorexigenic peptides in obese and non-obese children with PWS.

In conclusion, we observed an altered profile of circulating peptides regulating appetite—especially nesfatin-1 and spexin—in non-obese children with Prader-Willi syndrome during growth hormone treatment and reduced energy intake. In addition, nesfatin-1 concentrations are associated with leptin concentrations and BMI values in these patients. It seems that lower concentrations of spexin could affect the lipid profile in children with PWS. Changes in anorexigenic peptides may play a role in the etiology of metabolic disorders in Prader-Willi syndrome despite the applied therapy.

## Figures and Tables

**Figure 1 nutrients-15-01240-f001:**
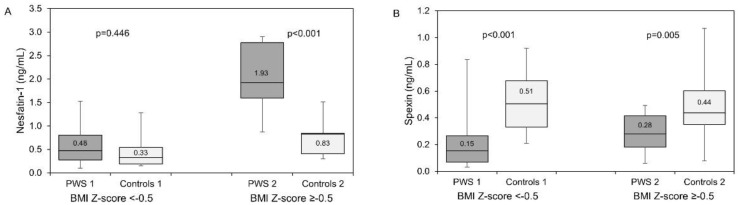
Comparison of nesfatin-1 (**A**), spexin (**B**), leptin (**C**), adiponectin and (**D**), concentrations between groups with PWS and the appropriate control groups. Patients with PWS (dark boxes), Control groups (bright boxes); BMI—body mass index.

**Table 1 nutrients-15-01240-t001:** Anthropometric and biochemical characteristics of children with PWS and healthy children.

Parameter	Children with PWS n = 25	Healthy Children n = 30	*p*-Value
Age (years)	6.3 ± 3.0	6.6 ± 3.0	0.709
Male (%)	48.0	46.7	0.921
**Anthropometric parameters**			
Height (cm)	115.5 ± 21.0	121.3 ± 20.5	0.299
Weight (kg)	21.0 ± 8.8	24.8 ± 9.5	0.132
BMI (kg/m^2^)	15.1 ± 1.5	16.2 ± 1.9	0.019
BMI Z-score	−0.58 (−1.19–−0.27)	−0.19 (−0.58–0.55)	0.011
Fat mass (%)	19.0 ± 4.6	20.9 ± 5.6	0.299
Fat mass (kg)	4.1 ± 2.2	5.2 ± 2.6	0.096
**Biochemical measurements**			
Nesfatin-1 (ng/mL)	0.91 (0.44–1.93)	0.65 (0.32–0.84)	0.019
Spexin (ng/mL)	0.20 (0.12–0.33)	0.45 (0.35–0.61)	<0.001
Leptin (ng/mL)	2.76 (1.25–3.95)	1.96 (0.98–3.98)	0.722
sOB-R (ng/mL)	43.2 ± 16.1	44.1 ± 15.7	0.753
Leptin/sOB-R	0.06 (0.03–0.11)	0.05 (0.02–0.09)	0.483
Total adiponectin (µg/mL)	11.5 ± 3.5	11.4 ± 4.4	0.574
Leptin/adiponectin	0.21 (0.1–0.39)	0.20 (0.08–0.38)	0.844
HMW adiponectin (µg/mL)	6.12 ± 2.27	7.04 ± 3.65	0.529
HMW adiponectin (%)	52.4 ± 8.4	60.6 ± 19.7	0.148
Proinsulin (pmol/L)	2.12 (1.04–3.47)	1.43 (0.91–2.18)	0.069
Glucose (mg/dL)	84.0 ± 6.2	84.0 ± 5.0	0.870
Total cholesterol (mg/dL)	172.2 ± 30.0	150.9 ± 20.6	0.001
HDL cholesterol (mg/dL)	57.5 ± 13.3	54.2 ± 12.9	0.251
LDL cholesterol (mg/dL)	110.0 ± 24.7	88.5 ± 16.4	<0.001
Triglycerides (mg/dL)	82.8 ± 26.5	65.1 ± 23.8	0.005
IGF-I (ng/mL)	307.5 (192.4–343.0)	143.5 (109.7–230.8)	<0.001
t-IGFBP-3 (µg/mL)	4.19 ± 1.27	3.41 ± 1.05	0.016
f-IGFBP-3 (µg/mL)	2.88 ± 1.35	2.22 ± 1.01	0.055
IGF-I/t-IGFBP-3 molar ratio	0.25 ± 0.07	0.17 ± 0.06	<0.001
f-IGFBP-3/t-IGFBP-3 molar ratio	0.67 ± 0.20	0.64 ± 0.18	0.394

The results are presented as means ± standard deviation (SD) for symmetric distributed data or medians and interquartile range (25–75th percentiles) for skewed distributed variables. BMI—body mass index; HMW adiponectin—high molecular weight adiponectin; sOB-R—soluble leptin receptor; HDL—high-density lipoprotein; LDL—low-density lipoprotein; IGF-I—insulin-like growth factor-I; t-IGFBP-3—total-IGF-binding protein-3; f-IGFBP-3—functional IGFBP-3.

**Table 2 nutrients-15-01240-t002:** Daily energy and nutrient intake of the examined children with PWS and the control group.

Parameter	Children with PWS n = 25	Healthy Children n = 30	*p*-Value
Energy (kcal/day)	1034 ± 316	1422 ± 424	0.001
Protein (g/day)	46.1 ± 17.9	44.7 ± 14.9	0.993
Carbohydrates (g/day)	143.7 ± 45.3	199.1 ± 67.3	0.001
Fat (g/day)	35.1 ± 14.2	53.4 ± 20.8	0.001
Cholesterol (mg/day)	128.6 (77.3–197.3)	205.1 (151.0–258.8)	0.006
Saturated fatty acids (g/day)	12.0 ± 5.7	19.7 ± 9.1	0.001
Fiber (g/day)	15.3 ± 5.9	14.2 ± 5.3	0.659
Energy (% of EER)	69.0 ± 14.0	88.6 ± 18.5	<0.001
Proteins (% of energy intake)	17.8 ± 4.0	13.0 ± 1.7	<0.001
Carbohydrates (% of energy intake)	50.7 ± 6.9	53.2 ± 6.6	0.132
Fat (% of energy intake)	29.8 ± 5.4	33.7 ± 5.6	0.018
Fiber (% of AI)	106.3 ± 35.3	91.8 ± 31.8	0.274

The results are presented as means ± standard deviation (SD) for symmetric distributed data or medians and interquartile range (25–75th percentiles) for skewed distributed variables. EER—estimated energy requirement; AI—adequate intake. Recommended daily energy and nutrient intakes (1–3/4–6/7–9/10–12 girls/10–12 boys, years) according to Jarosz [29]: energy (1000/1400/1800/2100/2350 kcal/day), protein (12/16/23/31/32 g/day), carbohydrates (130 g/day), fat (33–44/31–54/34–60/47–82/52–91 g/day), protein (0–2 years, 5–15%; 3–18 years, 10–20%), carbohydrates (1–18 years, 45–65%), fat (1–3 years 35–40%; 4–18 years, 20–35%), saturated fatty acids (11.1/15.6/20.2/11.8–14.1/13.1–15.8 g/day), and fiber (10/14/16/19/19 g/day).

**Table 3 nutrients-15-01240-t003:** Anthropometric and biochemical characteristics and dietary intake of PWS subgroups and their respective controls.

Parameter	Children with BMI Z-Score < −0.5			Children with BMI Z-Score ≥ −0.5			PWS1 vs. PWS2	Controls1 vs. Controls2
	PWS1 (n = 13)	Controls1 (n = 10)	*p*	PWS 2 (n = 12)	Controls 2 (n = 20)	*p*	*p*	*p*
Age (years)	5.5 ± 3.3	7.1 ± 2.8	0.162	7.1 ± 2.6	6.4 ± 3.2	0.624	0.164	0.657
Height (cm)	110.9 ± 23.4	127.0 ± 21.0	0.091	120.4 ± 17.7	118.5 ± 20.1	1.000	0.264	0.502
Weight (kg)	17.9 ± 7.8	24.4 ± 10.4	0.088	24.3 ± 9.0	24.9 ± 9.3	0.803	0.060	0.854
BMI (kg/m^2^)	14.1 ± 0.6	14.5 ± 1.4	0.512	16.2 ± 1.4	17.1 ± 1.6	0.078	<0.001	<0.001
BMI Z-score	−1.15 (−1.51–−0.75)	−0.89 (−1.16–−0.56)	0.119	−0.27 (−0.41–0.17)	0.28 (−0.2–0.74)	0.035	<0.001	<0.001
Fat mass (%)	17.2 ± 4.5	19.3 ± 2.9	0.171	20.9 ± 3.9	21.7 ± 6.4	0.916	0.031	0.455
Fat mass (kg)	3.2 ± 1.8	4.5 ± 1.9	0.101	5.0 ± 2.3	5.6 ± 2.9	0.716	0.060	0.588
**Biochemical measurements**								
Leptin/sOB-R	0.03 (0.01–0.06)	0.03 (0.02–0.07)	0.636	0.1 (0.08–0.17)	0.06 (0.03–0.1)	0.037	<0.001	0.072
sOB-R (ng/mL)	51.1 ± 18.3	49.3 ± 14.7	0.879	34.7 ± 6.7	41.6 ± 15.9	0.307	0.016	0.172
Leptin/adiponectin	0.11 (0.05–0.18)	0.13 (0.06–0.27)	0.343	0.35 (0.31–0.50)	0.24 (0.10–0.38)	0.060	<0.001	0.267
Total cholesterol (mg/dL)	159.2 ± 28.7	139.7 ± 17.8	0.029	186.3 ± 25.5	156.6 ± 20.0	0.001	0.054	0.036
HDL-cholesterol (mg/dL)	53.5 ± 12.4	57.8 ± 12.3	0.455	61.8 ± 13.3	52.5 ± 13.1	0.054	0.181	0.165
LDL-cholesterol (mg/dL)	101.9 ± 23.8	75.4 ± 15.8	0.008	118.7 ± 23.5	95.1 ± 12.6	0.003	0.126	0.001
Triglycerides (mg/dL)	77.2 ± 23.1	57.3 ± 14.4	0.024	89.0 ± 29.5	69.0 ± 26.8	0.054	0.327	0.379
IGF-I/t-IGFBP-3 molar ratio	0.25 ± 0.07	0.17 ± 0.05	0.008	0.25 ± 0.06	0.17 ± 0.06	0.002	0.799	0.838
**Dietary intake**								
Energy (kcal/day)	981 ± 300	1480 ± 379	0.004	1090 ± 337	1393 ± 451	0.076	0.470	0.619
Energy (% of EER)	72.5 ± 15.3	88.4 ± 22.4	0.085	65.3 ± 12.1	88.8 ± 17.0	<0.001	0.398	0.787
Protein (% of energy intake)	17.1 ± 4.1	13.7 ± 1.8	0.030	18.7 ± 4.0	12.6 ± 1.6	<0.001	0.283	0.093
Carbohydrates (% of energy intake)	51.8 ± 5.2	51.5 ± 7.4	0.962	49.4 ± 8.5	54.0 ± 6.1	0.142	0.865	0.454
Fat (% of energy intake)	28.9 ± 3.8	34.7 ± 6.2	0.049	31.0 ± 7.0	33.2 ± 5.4	0.333	0.691	0.481
Protein (g/day)	42.4 ± 17.6	45.9 ± 13.6	0.563	50.1 ± 18.2	44.0 ± 15.9	0.477	0.247	0.846
Carbohydrates (g/day)	139.8 ± 39.3	198.9 ± 76.5	0.067	148.0 ± 52.6	199.1 ± 64.4	0.009	0.810	1.000
Fat (g/day)	32.0 ± 12.4	57.5 ± 18.9	0.001	38.6 ± 15.8	51.3 ± 21.8	0.182	0.295	0.530
Cholesterol	94.4 (65.5–170.8)	228.4 (169.1–251.4)	0.015	147.2 (108.2–246.7)	202.4 (142.9–279.4)	0.207	0.077	0.892
Saturated fatty acids (g/day)	10.3 ± 4.7	20.4 ± 10.2	0.004	14.0 ± 6.1	19.2 ± 8.6	0.104	0.060	0.739
Fiber (g/day)	15.0 ± 4.9	12.7 ± 6.7	0.376	15.6 ± 7.0	15.1 ± 4.2	0.836	0.769	0.259
Fiber (% of AI)	111.3 ± 31.1	79.9 ± 43.7	0.049	100.9 ± 40.1	98.8 ± 20.7	0.556	0.650	0.040

The results are presented as means ± standard deviation (SD) for symmetric distributed data or medians and interquartile range (25–75th percentiles) for skewed distributed variables. BMI—body mass index; sOB-R—soluble leptin receptor; HDL—high-density lipoprotein; LDL—low-density lipoprotein; IGF-I—insulin-like growth factor-I, t-IGFBP-3—total-IGF-binding protein-3; EER—estimated energy requirement; AI—adequate intake.

**Table 4 nutrients-15-01240-t004:** Associations between peptides regulating appetite and selected anthropometric parameters in the entire group of children with PWS.

Parameter	Nesfatin-1			Spexin			Leptin			Adiponectin		
	Coefficient	*p*-Value	95% CI	Coefficient	*p*-Value	95% CI	Coefficient	*p*-Value	95% CI	Coefficient	*p*-Value	95% CI
BMI	0.344	0.018	0.065–0.623	−0.001	0.960	−0.061–0.059	0.934	0.001	0.408–1.460	−0.619	0.349	−1.962–.723
BMI Z-score	0.711	0.031	0.073–10.349	0.034	0.600	−0.098–0.166	1.473	0.027	0.180–2.767	−2.030	0.205	−5.253–1.192
Fat mass (%)	0.076	0.169	−0.035–0.188	0.002	0.808	−0.018–0.022	0.235	0.026	0.031–0.439	−0.028	0.909	−0.541–0.484
Nesfatin-1	-	-	-	0.011	0.793	−0.075–0.097	1.345	0.042	0.050–2.640	−0.679	0.666	−3.902–2.543
Spexin	−0.830	0.624	−4.297–2.637	-	-	-	−0.762	0.823	−7.731–6.208	−3.391	0.613	−17.116–10.334
Leptin	0.244	0.042	0.010–0.478	−0.005	0.805	−0.048–0.038	-	-	-	−0.296	0.657	−1.662–1.069
Adiponectin	0.045	0.583	−0.121–0.211	0.001	0.954	−0.024–0.025	−0.030	0.864	−0.392–0.332	-	-	-

BMI—body mass index; CI—confidence interval.

**Table 5 nutrients-15-01240-t005:** Nesfatin-1 and spexin concentrations in children with PWS and controls depending on the nutritional phase.

Group	Parameter	Phase 2a (20–31 Months)	Phase 2b (3–5.25 Years)	Phase 3 (6–12 Years)	*p* _trend_
Children with PWS n = 25	Nesfatin-1 (ng/mL)	0.40 (0.29–1.57) N = 3	0.30 (0.25–0.87) N = 7	1.74 (0.83–2.70) N = 15	0.004
Children with PWS n = 30	Spexin (ng/mL)	0.08 (0.06–0.17) N = 3	0.15 (0.06–0.33) N = 7	0.23 (0.17–0.35) N = 15	0.041
Healthy children n = 30	Nesfatin-1 (ng/mL)	0.61 (0.34–0.83) N = 4	0.40 (0.27–0.45) N = 6	0.83 (0.32–0.90) N = 20	0.132
Healthy children n = 30	Spexin (ng/mL)	0.50 (0.31–0.77) N = 4	0.44 (0.34–0.57) N = 6	0.44 (0.34–0.76) N = 20	0.698

The results are presented as medians and interquartile range (25–75th percentiles) for skewed distributed variables.

## Data Availability

The data presented in this study are available upon reasonable request to the corresponding author.

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
