# Peer review of "Circulating Levels of Nesfatin-1 and Spexin in Children with Prader-Willi Syndrome during Growth Hormone Treatment and Dietary Intervention"

_nutrients, 2023, doi:10.3390/nu15051240_

Round 1
Reviewer 1 Report
Summary:
This is a well-written, interesting and novel article in which the authors analyzed the profiles of anorexigenic peptides in patients with PWS (2-12 years) with growth hormone treatment and reduced energy intake. Altered profiles of Nesfatin-1 and Spexin were found in non-obese PWS patients vs healthy children.
Comments:
- P2, L1: please correct oxygenic, orexogenic?
- Please include the different nutrional phases in your article. Children and adolescents with PWS experience complex and multiple nutrional phases (e.g. PMID: 21465655), progressing from failure to thrive to hyperphagia and obesity through different stages. Neuropeptides might play different roles during these different nutritional phases. It would be interesting to perform a subanalysis in the different age groups to look for different expression of the (an)orexigenic peptides.
- Information on animal studies is lacking. Please add this information in the introduction and/or discussion of the article.
- Table 1: p-value of Nesfatin-1 is 0.019 in the table and <0.001 in the text; p-value of Spexin is <0.001 in the Table and p=0.019 in the text. Please correct.
- Table 2: split table (or add an horizontal line) between absolute and relative values
- Table 3: p-values between PWS-1 and PWS-2 are not shown in the table but are discussed in the text. This is very confusing.
- Is there an association of spexin or nesfatin-1 with the % of protein intake?
- Is there an age-dependent influence on the values of spexin and nesfatin-1 in PWS1 or PWS2?
- In the literature, highly conflicting results regarding nesfatin-1 and spexin were reported in both malnourished and obese children. This suggests that other factors or a combination of factors play a role. Are all values in literature reported after fasting? Does fasting versus non-fasting play a role? Age effect in PWS?
Author Response
Below are our comments regarding the reviewer’s remarks:
Reviewer 1:
Summary:
This is a well-written, interesting and novel article in which the authors analyzed the profiles of anorexigenic peptides in patients with PWS (2-12 years) with growth hormone treatment and reduced energy intake. Altered profiles of Nesfatin-1 and Spexin were found in non-obese PWS patients vs healthy children.
Comments:
- P2, L1: please correct oxygenic, orexigenic?
We corrected oxygenic to orexigenic.
- Please include the different nutrional phases in your article. Children and adolescents with PWS experience complex and multiple nutrional phases (e.g. PMID: 21465655), progressing from failure to thrive to hyperphagia and obesity through different stages. Neuropeptides might play different roles during these different nutritional phases. It would be interesting to perform a subanalysis in the different age groups to look for different expression of the (an)orexigenic peptides.
We added the following text to the Introduction section:
There are a number of nutritional phases in children and adolescents with PWS [5]. During phase 1, the infants are hypotonic and not obese. Phase 2 is associated with weight gain, but in sub-phase 2a the weight increases without a significant change in appetite or caloric intake (age quartiles 20-31 months). Next, in sub-phase 2b the weight gain is associated with a concomitant increased interest in food (quartiles 3-5.25 years). Phase 3 is characterized by hyperphagia accompanied by a lack of satiety (quartiles 5-13 years).
We added reference nr 5:
Miller, J.L.; Lynn, C.H.; Driscol,l D.C.; Goldstone, A.P.; Gold, J.A.; Kimonis, V.; Dykens, E.; Butler, M.G.; Shuster, J.J.; Driscoll, D.J. Nutritional phases in Prader-Willi syndrome. Am. J. Med. Genet. A. 2011,155A, 1040-1049.
We added the text into the 2.5. Statistical Analyses section:
For trend analysis, the Jonckheere-Terpstra test was used.
We added the following text and Table 5 to the Results section:
3.5 Nesfatin-1 and spexin concentrations in children with PWS depending on the nutritional phase
In children with PWS, we found significant associations between both neuropeptides and the nutritional phases (ptrend=0.004 for nesfatin-1; ptrend=0.041 for spexin) (Table 5).
Table 5. Nesfatin-1 and spexin concentrations in children with PWS and controls depending on the nutritional phase
|
Group
|
Parameter
|
Phase 2a (20-31 months)
|
Phase 2b (3-5.25 years)
|
Phase 3 (6-12 years)
|
ptrend
|
|
Children with PWS n=25 |
Nesfatin-1 (ng/mL) |
0.40 (0.29 – 1.57) N=3 |
0.30 (0.25 – 0.87) N=7 |
1.74 (0.83 - 2.70) N=15 |
0.004
|
|
Children with PWS n=30 |
Spexin (ng/mL) |
0.08 (0.06 – 0.17) N=3 |
0.15 (0.06 – 0.33) N=7 |
0.23 (0.17 – 0.35) N=15 |
0.041
|
|
Healthy children n=30 |
Nesfatin-1 (ng/mL) |
0.61 (0.34 – 0.83) N=4 |
0.40 (0.27 – 0.45) N=6 |
0.83 (0.32 – 0.90) N=20 |
0.132
|
|
Healthy children n=30 |
Spexin (ng/mL) |
0.50 (0.31 – 0.77) N=4 |
0.44 (0.34 – 0.57) N=6 |
0.44 (0.34 – 0.76) N=20 |
0.698
|
We observed the highest values of these peptides in the group with PWS aged 6-12 years (phase 3). In healthy children, we did not find any associations between the concentrations of nesfatin-1 and spexin and the nutritional phases.
We added the following text in the Discussion section:
“It seems that the higher nesfatin-1 concentrations may appear in response to the body’s energy status and/or nutritional phase. Patients belonging to the age group characterized by hyperphagia (phase 3) had the highest values of nesfatin-1 in comparison with nutritional phases 2a and 2b. Moreover, the patients with the higher BMI Z-score had more body fat mass, higher leptin concentrations, and a higher leptin/sOB-R ratio than children with lower BMI-Z-scores.”
“However, we found significant associations between spexin and the nutritional phases in our patients with PWS.”
- Information on animal studies is lacking. Please add this information in the introduction and/or discussion of the article.
We added the following text to the Introduction section:
The conservation of the PWS genetic interval on human chromosome 15q11-q13 and the gene cluster on mouse chromosome 7 facilitated the use of mice as animal models of PWS [2 ]. Some models mimicked the loss of all gene expression from the paternally inherited PWS genetic interval, while others considered smaller regions or single genes. These models revealed a number of mechanisms responsible for hypothalamic dysfunctions, resulting in hyperphagia, growth retardation and metabolic disorders.
We added reference nr 2:
Zahova, S.; Isles, A.R. Animal models for Prader-Willi syndrome. Handb. Clin. Neurol. 2021,181, 391-404.
- Table 1: p-value of Nesfatin-1 is 0.019 in the table and <0.001 in the text; p-value of Spexin is <0.001 in the Table and p=0.019 in the text. Please correct.
We corrected the p-values for nesfatin-1 and spexin.
- Table 2: split table (or add an horizontal line) between absolute and relative values
We changed Table 2 by adding an horizontal line between absolute and relative values.
- Table 3: p-values between PWS-1 and PWS-2 are not shown in the table but are discussed in the text. This is very confusing.
We modified Table 3 by adding columns with p-values for PWS1 vs PWS2 an Control1 vs Control2.
- Is there an association of spexin or nesfatin-1 with the % of protein intake?
There is a significant correlation between nesfatin 1 and % protein intake (p=0.035) in group with PWS, however after age adjustment in quantile regression the effect becomes nonsignificant (p>0.05).
- Is there an age-dependent influence on the values of spexin and nesfatin-1 in PWS1 or PWS2?
The results seem to be inconsistent because of the small size of the PWS1 and PWS2 subgroups. However, there are statistically significant correlations between nesfatin-1 (rho=0.558, p=0.004) and spexin (rho=0.445, p=0.026) and age in the whole group with PWS. There are statistically nonsignificant correlations between nesfatin-1 (rho=0.192, p=0.309) and spexin (rho=0.167, p=0.377) and age in the whole control group.
- In the literature, highly conflicting results regarding nesfatin-1 and spexin were reported in both malnourished and obese children. This suggests that other factors or a combination of factors play a role. Are all values in literature reported after fasting? Does fasting versus non-fasting play a role? Age effect in PWS?
We have added the following additions to the Discussion section regarding nesfatin-1 and spexin:
„Conflicting results were reported regarding fasting serum nesfatin-1 concentrations and the relation between nesfatin-1 and BMI values in malnourished as well as obese children [18-20, 37-39].”
„It also cannot be ruled out that nesfatin-1 synthesis and serum nesfatin-1 concentration may differ in the studied populations due to nesfatin-1 gene polymorphism, as suggested by some authors. [40].”
„It seems that the higher nesfatin-1 concentrations may appear in response to the body’s energy status and/or nutritional phase. Patients belonging to the age group characterized by hyperphagia (phase 3) had the highest values of nesfatin-1 in comparison with nutritional phases 2a and 2b.”
„ However, we found significant associations between spexin and the nutritional phases in our patients with PWS. In the hypothalamus, spexin signals neurons directly reducing food intake by enhancing leptin receptor and melanocortin 4 receptor expression while decreasing neuropeptide Y type 5 receptor, and ghrelin receptor expression [47]. Therefore, it is suggested that spexin expression is regulated by metabolic status or feeding conditions.”
We added references nr 40 and 47:
- Zegers D, Beckers S, Mertens IL, Van Gaal LF, Van Hul W. Association between polymorphisms of the Nesfatin gene, NUCB2, and obesity in men. Mol Genet Metab. 2011 Jul;103(3):282-6.
- Wong MKH, Chen Y, He M, Lin C, Bian Z, Wong AOL. Mouse Spexin: (II) Functional role as a ratiety factor inhibiting food intake by regulatory actions within the hypothalamus. Front Endocrinol (Lausanne). 2021 Jul 2;12:681647.
Reviewer 2 Report
The aim of this study was: to analyze the differences in the profiles of circulating peptides reg-ulating appetite-mainly nesfatin-1 and spexin in children with Prader-Willi syndrome (PWS) undergoing GH treatment and reduced energy intake with the profiles in healthy, non-obese children, and to evaluate the relationships between the biochemical parameters and anthropometric indices in children with PWS during GH treatment and dietary intervention. The study population consisted of 25 non-obese children with PWS and 30 healthy children of the same age following an unrestricted age-energy intake.
In the present study, the authors observed an altered profile of circulating peptides regulating appetite especially nesfatin-1 and spexin in non-obese children with PWS during growth hormone treatment and reduced energy intake. Higher concentrations of nesfatin-1 in children with PWS than in healthy children and positive associations between nesfatin-1 and BMI and BMI Z-score were found. The positive associations between nesfatin-1 and leptin in children with PWS was also found. Moreover, children with PWS had lower serum spexin concentrations compared with the controls. Significant differences in lipid profile between those groups were also observed.
The authors’ conclusion was that changes in anorexigenic peptides could play a role in the etiology of metabolic disorders in PWS despite the applied therapy.
The exact mechanism of obesity development in Prader-Willi syndrome is still not fully understood.
The present study is innovative and is the first to evaluate anorexigenic neuropeptides nesfatin-1 and spexin levels in relation with anthropometric parameters and other peptides regulating appetite in children with PWS.
The purpose of the study is interesting. The results of the study are interesting from the scientific and perhaps in the future also from the practical point of view.
From the methodological point of view that Paper has many strong points:
1. Appropriate and innovative design of the study
2. Clear and logical presentation of the results
3. The tables and figures presented the results concisely and clearly
4. Clear inclusion and exclusion criteria
5. Good statistical data processing
6. Interesting conclusions of the study.
Although the present study had some limitations as: relatively small number of participants owing to the rarity of PWS; cross-sectional nature and the absence of a prospective longitudinal analysis; the lack of comparison between non-obese and obese children with PWS, these limitations do not diminish its scientific value.
I have only one comment for the Authors: correctly use the abbreviation PWS throughout the text of the paper.
Author Response
Below are our comments regarding the reviewer’s remarks:
Reviewer 2:
I have only one comment for the Authors: correctly use the abbreviation PWS throughout the text of the paper.
We have used the abbreviation PWS correctly throughout the text.
Reviewer 3 Report
The manuscript by Joanna Gajewska and colleagues analyses the levels of neuroendocrine peptides regulating appetite, lipid profile and IGF1 in a cohort of 25 children with Prader Willi syndrome treated with growth hormone therapy and energy-restricted diet.
The levels of the anorexigenic peptides nesfatin-1 and spexin have been poorly investigated in Prader-Willy syndrome and hence the study provides original data.
Taking into account that Prader Willi syndrome is a rare disorder, the total number of subjects analysed can be considered fair.
The manuscript is globally well presented, the methods clearly described and this reviewer suggests to consider the paper for publication after the following minor revisions:
- Introduction
Page 2, line 1: the authors use the term “oxygenic”. Do they mean “oxygenic” or “orexigenic”?
- Methods:
Page 3. The sentence “The control group consisted of 30 non-obese, healthy children
(BMI Z-score <-1 + 1>) within the same age range as the obese group with an adequate
nutritional or dietary status” should be better clarified: 1. Z-score <-1 + 1> means a z-score between -1 and 1? It is unclear; 2. What is the “obese group” the authors refer to? In the paper there is no obese group taken into consideration…please clarify.
- Results.
Table 1.
1. the authors compare children with PWS with healthy children of the same age. PWS patients are treated with GH therapy for at least 1 year (but the authors do not specify the total number of years of treatment/patient). It is impressive to note from the table that the height in PWS is not statistically different from healthy subjects. I suggest to add a table (also as supplementary table) with the age, gender, molecular diagnosis, anthropometric data and total number of years of GH therapy for the 25 PWS patients. I believe that these data will be interesting to readers.
2. The numbers indicated in brackets refer to interquartile ranges? This must be added to table footnote.
- Discussion:
1. Do the authors think that the findings of this study could influence the management of patients with PWS? Will these results help in clinical practice? I encourage the authors to express their opinion in the discussion section.
2. Please change “Prader’s syndrome” with “Prader-Willi syndrome”
Author Response
Below are our comments regarding the reviewer’s remarks:
Reviewer 3:
The manuscript is globally well presented, the methods clearly described and this reviewer suggests to consider the paper for publication after the following minor revisions:
- Introduction
Page 2, line 1: the authors use the term “oxygenic”. Do they mean “oxygenic” or “orexigenic”?
We corrected oxygenic to orexigenic.
- Methods:
Page 3. The sentence “The control group consisted of 30 non-obese, healthy children (BMI Z-score <-1 + 1>) within the same age range as the obese group with an adequate nutritional or dietary status” should be better clarified: 1. Z-score <-1 + 1> means a z-score between -1 and 1? It is unclear; 2. What is the “obese group” the authors refer to? In the paper there is no obese group taken into consideration…please clarify.
It was our mistake. We changed the sentence:
„The control group consisted of 30 non-obese, healthy children (BMI Z-score <-1 + 1>) within the same age range as the group with PWS, an adequate nutritional or dietary status according to the recommendations of Kułaga et al. [28] and Jarosz et al. [29].”
- Results.
Table 1.
- the authors compare children with PWS with healthy children of the same age. PWS patients are treated with GH therapy for at least 1 year (but the authors do not specify the total number of years of treatment/patient). It is impressive to note from the table that the height in PWS is not statistically different from healthy subjects. I suggest to add a table (also as supplementary table) with the age, gender, molecular diagnosis, anthropometric data and total number of years of GH therapy for the 25 PWS patients. I believe that these data will be interesting to readers.
We added the following sentence to the Materials and methods section, Patients:
“The average duration of treatment with growth hormone in the whole group of patients with PWS was 4.7 ± 2.8 years.”
We agree that examining the relationships between anthropometric and biochemical parameters, including new neuropeptides, and genetic data will be of great interest for assessing the clinical utility of these peptides in patients with PWS. This will be the subject of our next study.
- The numbers indicated in brackets refer to interquartile ranges? This must be added to table footnote.
We added the following text to table footnote:
“The results are presented as means ± standard deviation (SD) for symmetric distributed data or medians and interquartile range (25th–75th percentiles) for skewed distributed variables.”
- Discussion:
- Do the authors think that the findings of this study could influence the management of patients with PWS? Will these results help in clinical practice? I encourage the authors to express their opinion in the discussion section.
Nesfatin-1 and spexin have not yet been studied in PWS patients. Our research on these peptides is the first in this group of children. Presented study was its cross-sectional nature and the absence of a prospective longitudinal analysis, which is needed to examine the relationship between circulating peptides regulating appetite and clinical outcomes in these subjects during therapy. Therefore, the assessment of the clinical utility of nesfatin-1 and spexin in patients with PWS requires long-term monitoring of the concentrations of these peptides during growth hormone therapy as well as dietary intervention.
We added the following sentence to Limitations:
„Therefore, the assessment of the clinical utility of nesfatin-1 and spexin in patients with PWS requires long-term monitoring of the concentrations of these peptides during growth hormone therapy as well as dietary intervention.”
- Please change “Prader’s syndrome” with “Prader-Willi syndrome”
We changed “Prader’s syndrome” with “Prader-Willi syndrome”.